# Prevalence and Distribution of HPV Genotypes in Immunosuppressed Patients in Lorraine Region

**DOI:** 10.3390/v13122454

**Published:** 2021-12-07

**Authors:** Margot Boudes, Véronique Venard, Thierry Routiot, Marie Buzzi, Floriane Maillot

**Affiliations:** 1Department of Medical Gynecology, CHRU Nancy, 10 rue Heydenreich, 54000 Nancy, France; f.maillot@chru-nancy.fr; 2Faculty of Medicine, University of Lorraine, 9 Avenue de la Forêt de Haye, 54505 Nancy, France; v.venard@chru-nancy.fr (V.V.); m.buzzi@chru-nancy.fr (M.B.); 3Laboratory of Virology, CHRU Nancy, 54505 Nancy, France; 4Department of Obstetrics & Gynecology, CHRU Nancy, 10 rue Heydenreich, 54000 Nancy, France; t.routiot@chru-nancy.fr

**Keywords:** human papillomavirus, HPV genotypes, immunosuppressed patients, HIV, immunosuppressive therapy

## Abstract

Background: The primary objective of this work was to assess the prevalence and distribution of HPV genotypes in immunosuppressed patients, and to compare them with the French Monsonego cohort. Secondary objectives were to evaluate whether the risk of HPV infection was correlated with HIV viral load, CD4 cell count in HIV-infected patients and the type, number of immunosuppressive therapies or type of pathology (transplant vs. autoimmune diseases) in patients undergoing long-term immunosuppressive therapy. Methods: An observational, monocentric and historical study was conducted including all immunosuppressed patients having received an HPV testing, in the Laboratory of Virology, Nancy Regional Teaching Hospital Center, between 2014 and 2020. Immunosuppressed patients were either HIV-infected or received long-term immunosuppressive therapy. Results: In our cohort, the prevalence of HPV infection (75.6% vs. 16.1% *p* < 0.05), the proportion of patients with high-risk HPV infection (48.9% vs. 15.1% *p* < 0.05) and with multiple HPV infection (41.1% vs. 5.7% *p* < 0.05) were significantly higher than in the Monsonego cohort. HPV 52 (13%), 53 (13%) and 16 (10%) were the most common in the immunosuppressed population, while it was HPV 16, 42 and 51 in the Monsonego cohort. Conclusions: This study supports that a particular attention must be given to all the immunosuppressed patients for the screening and care of HPV-related diseases because of major modifications of HPV epidemiology compared with the overall population.

## 1. Introduction

Immunosuppressed population includes HIV-infected patients, transplant patients and those affected by long-term immunosuppressive therapy for autoimmune diseases. In France, 173,000 people are living with HIV (human immunodeficiency virus) [1]. In 2018, after several years of stagnation a slight decrease was noted in the number of people discovering their seroconversion [1]. However, the number of transplants in France has regularly increased, notably between 2018 and 2019 with a rise of 1.6% [2]. Sixty-thousand people are currently living with a transplanted organ in France. Ten percent of industrialized countries populations are affected by autoimmune diseases and this number has been constantly rising over the last few decades [3].

The immunosuppressed population has an increased risk of infectious and tumoral complications compared with the healthy population. Concerning HPV (human papillomavirus) infection, the reduction in immune defenses causes a reactivation of latent HPV infection with a decrease of viral clearance responsible of the extended persistence of the virus. Therefore, the prevalence of HPV infection is higher among the immunosuppressed population. For this reason, benign lesions, dysplasias and cancers due to HPV are more common in immunosuppressed patients [4].

A French study demonstrated that abnormal cervical smears in screening were much more frequent in HIV-infected women than in the overall population: 42% vs. 5%, *p* < 0.001 [5]. According to a German study, tritherapy had no impact on the occurrence of cervical cancer [6]. In two American studies, the risk of cervical cancer increased from a factor of 2 to 22 in the presence of HIV infection and this risk depended on CD4 cell count [4,7].

Concerning population with a long-term immunosuppressive treatment, Grulich et al. reported a higher standardized incidence ratio (SIR) of HPV-associated cancers in transplant patients compared to the overall population [8]. Other studies showed a 3.5 to 5-fold increased risk for cervical cancer in transplant populations [9,10]. Some studies also described an increased risk of HPV-related diseases in patients affected by autoimmune diseases such as systemic lupus erythematosus [11,12] or inflammatory bowel disease [13,14].

A large number of studies have described the epidemiology of HPV infection in HIV-infected patients [15,16,17,18], but none on the French population. Among patients with a long-term immunosuppressive therapy, most studies were carried out on women with renal transplant [19,20]. The object of this work was to assess prevalence and distribution of HPV genotypes in a large, immunosuppressed population (HIV-infected patients and patients with a long-term immunosuppressive therapy, including transplant patients as well as those with autoimmune diseases) and to compare them with the French cohort described by Monsonego [21]. In Monsonego’s study, 5002 women were included when they consulted for a routine check in Paris region, France. We selected this population because it seemed to be the most reliable and representative data of the French overall population. Moreover, most studies which described the distribution of HPV genotypes in immunosuppressed populations only specified high-risk HPV genotypes. The secondary objectives were to assess if the risk of HPV infection was correlated with HIV viral load, CD4 cell count in HIV-infected patients and the type, number of immunosuppressive therapies or type of pathology in patients undergoing long-term immunosuppressive therapy. To our knowledge, this is one of the most recent studies in the Lorraine region because it did not participate in the French HPV genotyping program with the EDiTH studies [22].

## 2. Methods

### 2.1. Population

All immunosuppressed women who received genital HPV genotyping at the Laboratory of Virology of Nancy Regional Teaching Hospital Center between 2014 and 2020 were eligible for inclusion (N = 98). We planned to exclude patients who consulted anonymously in family planning or who received an HPV genotyping other than genital. There was no age limit, and vaccination status was not an exclusion criterion. However, we only included patients who were screened with the INNO-LiPA^®^ HPV Genotyping Extra II kit from Fujirebio society (N = 90).

### 2.2. Data Collection

A nominative list of patients who received a genital HPV genotyping in the Laboratory of Virology of the Nancy Regional Teaching Hospital Center from 2014 to 2020 has been established. Among these patients, 98 were immunosuppressed and 90 met inclusion criteria (Figure 1). These HPV tests were realized either for screening or after an abnormal cytological or histological result. Data were collected from the Dxcare medical records in the Department of Gynecology of the Nancy Regional Teaching Hospital Center. The data collection was anonymous and was presented in an Excel software spreadsheet for statistical analysis. Collected data were sociodemographic, medical, virological, cytological and histological. Medical data covered gyneco-obstetrical and vaccine characteristics, as well as immunodepression and drug consumption. All genotyping was carried out at the Laboratory of Virology of the Nancy Regional Teaching Hospital Center using the INNO-LiPA^®^ HPV Genotyping Extra II kit from Fujirebio society; technical genotyping consisted of PCR SFP 10 with hybridization on a strip. The kit tested for 32 HPV genotypes:

Thirteen high-risk HPV (HR): 16, 18, 31, 33, 35, 39, 45, 51, 52, 56, 58, 59, 68.

Six HPV potentially high-risk (HR): 26, 53, 66, 70, 73, 82.

Thirteen low-risk HPV (LR): 6, 11, 40, 42, 43, 44, 54, 61, 62, 67, 81, 83, 89.

### 2.3. Data Analysis

We performed a descriptive analysis of the sample (mean and standard deviation for quantitative variables; headcount and percentage for qualitative variables). We compared the prevalence of various types of HPV infection in the immunosuppressed population with those in the Monsonego cohort [21] using a Chi^2^ test (or a Fisher’s test if headcounts were insufficient). We used the software R v4.0.3 and its interface R Studio.

### 2.4. Ethics

This study was approved by the Delegation for Clinical Research and Innovation, Nancy, under the number 2020PI067.

## 3. Results

### 3.1. Characteristics of the Cohort

The study involved 90 patients (median age: 44.2 years), 29/90 (32.2%) were HIV-infected patients and 61/90 (67.8%) received a long-term immunosuppressive therapy. 68/90 patients (75.6%) had regular gynecological follow up, which meant at least one consultation per year. Only 2/90 patients (2.2%) were vaccinated for HPV with Gardasil^®^. 22/90 patients (24.4%) reported active smoking. Among sexually transmitted infections other than HIV, 9/90 patients (10%) had a hepatitis B active infection, 1/90 patient (1.1%) a hepatitis C active infection and 1/90 patient (1.1%) had syphilis. The characteristics of the cohort are described in Table 1.

In HIV-infected population (N = 29), all women (29/29, 100%) were under antiretroviral treatment but 7 of 29 (24.1%) still presented a positive viral load (range 13, 9 × 10^4^ copies/mL, mean 4.6 × 10^4^ copies/mL) quantified by an Aptima HIV-1 Quant Dx test, Hologic, France. All the characteristics of the HIV-infected population are described in Table 2.

In the population with long-term immunosuppressive therapy (N = 61), 17/61 patients (27.9%) had a solid organ transplant, 5/61 patients (8.2%) bone marrow transplant and 39/61 patients (63.9%) an autoimmune disease, 9/61 patients (14.7%) had an inflammatory bowel disease, 7/61 patients (11.5%) systemic lupus erythematosus, 20/61 patients (32.8%) an inflammatory arthritis and 3/61 patients (4.9%) a multiple sclerosis. Monotherapy was received by 37/61 patients (60.7%) and 24/61 patients (39.3%) received combination therapy. Among patients receiving monotherapy, 9/37 patients (24.3%) received glucocorticoid therapies and 28/37 (75.7%) received non-glucocorticoid therapies. Immunodepression lasted 8.5 years on average (Table 3).

### 3.2. Distribution of Cytological/Histological Results

The distribution of cytological results was as follows (Table 4): 12 patients (12/90, 13.3%) had a normal cytology, 46 patients (46/90, 51.1%) ASC-US, 8 patients (8/90, 8.9%) ASC-H, 20 patients (20/90, 22.2%) LSIL, 1 patient (1/90, 1.1%) HSIL, 1 patient (1/90, 1.1%) AGC and 1 patient (1/90, 1.1%) AGUS/AIS.

Concerning histological results (Table 4), 19 patients (19/43, 44.2%) had a normal histology, 13 patients (13/43, 30.2%) CIN1, 3 patients (3/43; 7.0%) CIN2, 6 patients (6/43, 14%) CIN3, 1 patient (1/43, 2.3%) VIN2 and 1 patient (1/43, 2.3%) VIN3. After the colposcopy, monitoring was decided for 26 patients (26/47, 55.3%) and treatment for 20 patients (20/47, 42.6%).

### 3.3. Prevalence of HPV Infection

The prevalence of HPV infection was 75.6% (68/90) in our immunosuppressed population: 86.2% for HIV-infected patients (25/29) and 70.5% for patients receiving a long-term immunosuppressive therapy (43/61).

The proportion of patients with multiple HPV infections was 41.1% (37/90) in our immunosuppressed population: 55.2% for HIV-infected patients (16/29) and 34.4% for patients receiving a long-term immunosuppressive therapy (21/61).

### 3.4. Distribution of HPV Genotypes and Cytological/Histological Diagnosis

The prevalence of HPV 16/18 infection increased with the severity of cytological (4/46 8.7% in ASC-US, 4/20 20% in LSIL, 2/9 22% in HSIL + ASC-H) and histological lesions (2/19 11% when there was no lesion, 2/13 15% in CIN1 and 2/9 22% in CIN2/3).

### 3.5. HPV Genotyping

The proportion of patients with high-risk HPV infection was 48.9% (44/90): 51.7% (15/29) and 47.5% (29/61) for HIV-infected patients and for patients receiving a long-term immunosuppressive therapy, respectively. The distribution of all the HPV genotypes analyzed, including co-infections, is presented in Table 5. In HIV-infected patients, HPV 61 (5/29, 17%), 16 (4/29, 14%) and 52 (4/29, 14%) were the most representative. HPV 53 (9/61, 15%), 52 (8/61, 13%) and 16 (5/61, 8.2%) were the most common in patients receiving long-term immunosuppressive therapy. We assessed the distribution of high-risk HPV (HRHPV) genotypes according to the type of immunosuppressive disease in Table 6 and according to the number of immunosuppressive therapies in Table 7.

### 3.6. Factors Associated with HPV Infection

We studied the risk of HPV infection according to HIV viral load, CD4 cell count and to monotherapy/combination therapy status (Table 8). We also assessed the risk of HPV infection according to the type of disease (transplanted/autoimmune diseases) and to the type of immunosuppressive therapy (only patients with monotherapy have been included), which are shown in Table 9.

## 4. Discussion

In this study of 90 patients recruited over a seven-year period, the prevalence of HPV infection reached 75.6% compared to 16.1% in the French Monsonego cohort (*p* < 0.05). This French cohort was the most representative of the overall French population because a large number of women (5002) were included after consulting for an undergoing routine check. Moreover, the characteristics of this population were close to those of our immunosuppressed population: no women were vaccinated (only 2.2% in our study), 10.3% of the patients were less than 25 years old (6.7% in our study) and 23.5% of the patients smoked (24.4% in our study). However, the results were difficult to compare because the HPV PCR methods were different. Monsonego’s study used the PapilloCheck genotyping kit, which tested for 25 HPV genotypes: 13 high-risk HPV and 12 other HPV genotypes. In the Fujirebio HPV genotyping used in our study, 32 HPV genotypes were tested: the same 13 high-risk genotypes analyzed by PapilloCheck and 19 other HPV genotypes not included in PapilloCheck. As we described in our cohort, HPV 52, 53 and 16 were the most common genotypes, whereas it was HPV 16, 51 and 42 in the Monsonego cohort. HPV 16, 52 and 53 were also looked for with PapilloCheck genotyping kit used in Monsonego’s study and HPV 16, 42 and 51 were also looked for with Fujirebio HPV genotyping used in our study, which limits the bias previously described. The proportion of multiple HPV infection (41.1% vs. 5.7% *p* < 0.05) and of high-risk HPV infection (48.9% vs. 15.1% *p* < 0.05) were significantly higher in immunosuppressed population than in Monsonego’s cohort.

The prevalence of HPV infection in HIV-infected patients was 44.8% to 91.2% according to the literature [16,17,18]. In our HIV-infected population, the prevalence of HPV infection was 86.2% and was significantly higher than in Monsonego’s cohort (*p* < 0.05). The proportion of patients with multiple HPV infection was 55.2% and was also significantly more important than in Monsonego’s cohort (*p* < 0.05). In other studies, this proportion of co-infection reached 14.3–27.8% [17,18]. In our study, the proportion of high-risk HPV infection was 51.7% and was also significantly higher than in Monsonego’s cohort (*p* < 0.05). This result was in concordance with the literature, where the proportion of high-risk HPV infection reached 41.0% [17].

The study of a large cohort of renal-transplant women in Holland demonstrated that the prevalence of HPV infection was 27.1%, whereas in the overall population it was 9 to 10%, 17.4% presented a high-risk HPV infection and 27.1% a multiple HPV infection [19]. In our study, among patients with long-term immunosuppressive therapy, the prevalence of HPV infection (70.5% *p* < 0.05), proportion of multiple HPV infection (34.4% *p* < 0.05) and of high-risk HPV infection (34.4% *p* < 0.05) were significantly higher than in Monsonego’s French cohort. These results are in concordance with another study that included patients affected by systemic lupus erythematosus [12].

Prevalence of HPV infection (86.2% vs. 70.5% *p* = 0.11), proportion of high-risk HPV infection (51.7% vs. 34.4% *p* = 0.06), of multiple HPV infection (55.2% vs. 34.4% *p* = 0.71) and of HPV 16 infection (14% vs. 8.2% *p* = 0.41) were higher in HIV-infected population than in population with a long-term immunosuppressive therapy even if the results were not significant. Thus, we supposed that patients with a long-term immunosuppressive therapy have a lower risk of HPV-related diseases than HIV-infected patients even if a close monitoring stays necessary. Other studies to evaluate in detail epidemiology of HPV and prevalence of HPV-related diseases in population with a long-term immunosuppressive therapy are clearly necessary.

In our work, the prevalence of HPV 16/18 infection increased with the severity of cytological and histological lesions. Our data confirmed those from the French studies EDiTH I, II and III, which showed that the prevalence of HPV 16 and 18 increased with the severity of lesion reaching 73% in invasive cancers [22].

HPV 61 (17%), 52 (14%) and 16 (14%) were the most representative genotypes in our HIV-infected population (Table 5). These results are in accordance with an American study in which HPV 16 was not the most frequent genotype [15]. In another study including 12 African countries, HPV 16 was the most frequently detected genotype in HIV-infected patients [18].

In our HIV-infected population, when we compared the groups CD4 count ≤ 349 c/mm^3^ (N = 11) and CD4 count > 350 c/mm^3^ (N = 18), the prevalence of HPV infection was close in both groups (82% vs. 89% *p* = 0.59). Nevertheless, the proportion of multiple HPV infection (73% vs. 44% *p* = 0.14) and of high-risk HPV infection (73% vs. 39% *p* = 0.08) were higher in the group CD4 count ≤ 349 c/mm^3^ than in the group CD4 count > 350 c/mm^3^, even if the differences were not significant probably because the headcounts were low. There was not a significant difference between the groups positive viral load and negative viral load, but we note that the headcounts of these groups were also low.

In the population undergoing long-term immunosuppressive therapy, HPV 53 (15%), 52 (13%) and 16 (8.2%) were the most common genotypes. These results are in accordance with a Danish study, including women who underwent renal or bone marrow transplants, where HPV 16 was not dominant. HPV 45 was the most common genotype (3.3%) and only one patient tested positive for HPV 16 (1.7%) [20]. Nevertheless, in other European studies, including renal-transplant women and patients undergoing immunosuppressive therapies for at least 3 months, HPV 16 was dominant [19,23].

In the population with long-term immunosuppressive therapy, the risk of HPV infection, multiple HPV infection and high-risk HPV infection were not significantly different between the groups undergoing monotherapy (N = 37/61) and combination therapy (N = 24/61). Nor was it significantly different between the groups with autoimmune diseases (N = 39/61) and transplant (N = 22/61). Only a few studies specifically examined the association between the type of immunosuppressive therapy and prevalence of HPV infection/HPV-related diseases [24,25,26]. We explored the risk of HPV infection according to the type of immunosuppressive therapy but found no significant difference between the groups using glucocorticoid therapies (N = 9/37) and non-glucocorticoid therapies (N = 28/37). All these groups, however, had low headcounts and more studies are necessary to explore these questions.

According to several previous studies [27,28,29], we confirmed that active smoking has a major impact on the epidemiology of HPV. The prevalence of HPV infection (82% vs. 74% *p* = 0.4), the proportion of multiple HPV infection (45% vs. 40% *p* = 0.6) and of high-risk HPV infection (55% vs. 47% *p* = 0.5) were higher in the tobacco group than in the non-tobacco group. Even if the results were not significant, it seems to show that active smoking accentuates immunosuppression in these immunosuppressed patients, like in the overall population.

As only 2 patients of 90 (2.2%) were vaccinated, and with the quadrivalent vaccine, we explored the potential impact of nonavalent vaccine in our cohort by assessing the prevalence of nine HPV genotypes included in the nonavalent vaccine (HPV 6, 11, 16, 18, 31, 33, 45, 52 and 58) according to cytological and histological lesions. These HPV genotypes were present in most of cytological (24/46 52% in ASCUS, 9/20 45% in LSIL and 4/9 44% in HSIL + ASC-H) and histological lesions (8/13 62% in CIN1, 5/9 56% in CIN2/3 and 2/2 100% in VIN2/3). Thus, we supposed that the nonavalent vaccine could offer to immunosuppressed patients an efficient protection against HPV-related diseases. The immunogenicity of HPV vaccine has been evaluated in several studies [30,31]. Although limited by small headcounts, these studies showed that a three-dose schedule of HPV vaccine permitted long-term immunogenicity in immunocompromised children. According to our study, the nonavalent vaccine could be efficient in these immunosuppressed patients to decrease prevalence of HPV infection and of HPV-related diseases. Of course, further studies are necessary to demonstrate this.

## Figures and Tables

**Figure 1 viruses-13-02454-f001:**
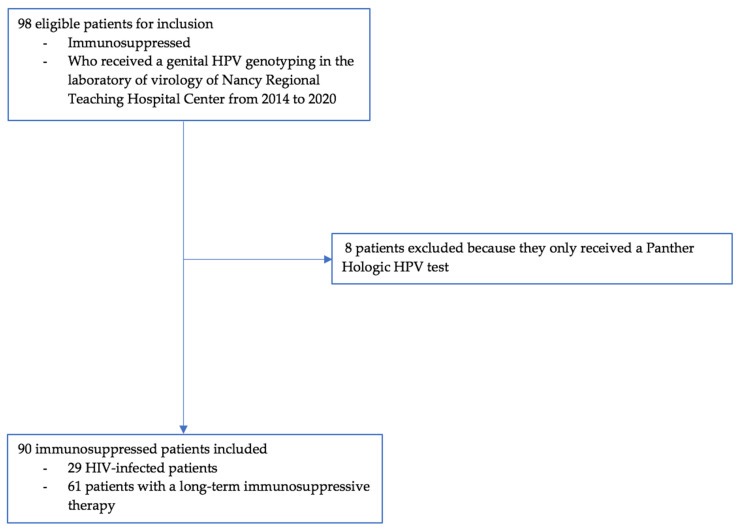
Flow chart.

**Table 1 viruses-13-02454-t001:** Characteristics of the population.

Characteristics	N = 90 ^1^
Age	44.2 (13.0)
Age < 25 years old	6 (6.7%)
Regular gynecological follow up *(one consultation per year)*	68 (75.6%)
Anti-HPV vaccination status	
Not vaccinated	88 (97.8%)
Gardasil	2 (2.2%)
Active smoking	
No	68 (75.6%)
Yes < 15 cigarettes a day	9 (10%)
Yes > 15 cigarettes a day	13 (14.4%)
Sexually transmitted infection (STI) other than HIV	
No	79 (87.8%)
Hepatitis B	9 (10%)
Hepatitis C	1 (1.1%)
Syphilis	1 (1.1%)
Immunodepression	
HIV	29 (32.2%)
Immunosuppressive therapy	61 (67.8%)

^1^ Mean (SD); n (%).

**Table 2 viruses-13-02454-t002:** Characteristics of the HIV-infected population.

Characteristics	N = 29 ^1^
HIV viral load ^2^	
Negative	22 (75.9%)
Positive (range 13.9 × 10^4^ copies/mL, mean 4.6 × 10^4^ copies/mL)	7 (24.1%)
Age at diagnosis of seropositivity *(years old)*	35.4 (10.2)
Current tritherapy	29 (100%)
Cumulative duration of tritherapy *(years)*	8.2 (7.7)
CD4 count over the last year	
≤349 c/mm^3^	11 (37.9%)
>350 c/mm^3^	18 (62.1%)

^1^ Mean (SD); n (%) ^2^ quantified by Aptima HIV-1 Quant Dx assay HOLOGIC, France; limit of detection 20 copies/mL.

**Table 3 viruses-13-02454-t003:** Characteristics of the population with long-term immunosuppressive therapy.

Characteristics	
Immunosuppressive diseases	N = 61 ^1^
Renal transplant	12 (19.7%)
Transplant of bone marrow	5 (8.2%)
Inflammatory bowel diseases	9 (14.7%)
Systemic lupus erythematosus	7 (11.5%)
Inflammatory arthritis	20 (32.8%)
Multiple sclerosis	3 (4.9%)
Pulmonary transplant	1 (1.6%)
Hepatic transplant	2 (3.3%)
Hepatic + renal transplant	2 (3.3%)
Number of immunosuppressive therapies	
1	37 (60.7%)
2	13 (21.3%)
3	11 (18%)
Monotherapies	37 ^1^
Glucocorticoids	9 (24.3%)
Non Glucocorticoids	28 (75.7%)
Cumulative duration of immunodepression *(years)*	8.5 (8.4)

^1^ Mean (SD); n (%).

**Table 4 viruses-13-02454-t004:** Distribution of cytological and histological results.

Cytology	N = 90 ^1^
Normal	12 (13.3%)
ASCUS	46 (51.1%)
ASC-H	8 (8.9%)
LSIL	20 (22.2%)
HSIL	1 (1.1%)
AGC	1 (1.1%)
AGUS/AIS	1 (1.1%)
Unknown	1 (1.1%)
Histology	N = 43 ^1^
No lesion	19 (44.2%)
CIN1	13 (30.2%)
CIN2	3 (7.0%)
CIN3	6 (14%)
VIN3	1 (2.3%)
VIN2	1 (2.3%)

^1^ Mean (SD); n (%). ASCUS = atypical squamous cells of unknown significance. ASC-H = atypical squamous cells cannot exclude HSIL. LSIL = low-grade squamous intraepithelial lesion. HSIL = high-grade squamous intraepithelial lesion. AGC = atypical glandular cells. AGUS/AIS = atypical glandular cells of undetermined significance/endocervical adenocarcinoma in situ. CIN = cervical intraepithelial neoplasia. VIN = vulvar intraepithelial neoplasia.

**Table 5 viruses-13-02454-t005:** Distribution of HPV genotypes.

HPV Genotypes	Immunosuppressed Population N = 90 ^1^	Immunosuppressive Therapy N = 61 ^1^	HIV N = 29 ^1^
0	1 (1.1%)	0 (0%)	1 (3.4%)
6	6 (6.7%)	3 (4.9%)	3 (10%)
11	0 (0%)	0 (0%)	0 (0%)
16	9 (10%)	5 (8.2%)	4 (14%)
18	4 (4.4%)	2 (3.3%)	2 (6.9%)
26	0 (0%)	0 (0%)	0 (0%)
31	7 (7.8%)	4 (6.6%)	3 (10%)
33	5 (5.6%)	3 (4.9%)	2 (6.9%)
35	5 (5.6%)	3 (4.9%)	2 (6.9%)
39	3 (3.3%)	2 (3.3%)	1 (3.4%)
40	1 (1.1%)	0 (0%)	1 (3.4%)
42	0 (0%)	0 (0%)	0 (0%)
43	1 (1.1%)	1 (1.6%)	0 (0%)
44	3 (3.3%)	3 (4.9%)	0 (0%)
45	3 (3.3%)	1 (1.6%)	2 (6.9%)
51	3 (3.3%)	2 (3.3%)	1 (3.4%)
52	12 (13%)	8 (13%)	4 (14%)
53	12 (13%)	9 (15%)	3 (10%)
54	2 (2.2%)	2 (3.3%)	0 (0%)
56	3 (3.3%)	2 (3.3%)	1 (3.4%)
58	5 (5.6%)	4 (6.6%)	1 (3.4%)
59	4 (4.4%)	4 (6.6%)	0 (0%)
61	6 (6.7%)	1 (1.6%)	5 (17%)
62	3 (3.3%)	3 (4.9%)	0 (0%)

^1^ Mean (SD); n (%).

**Table 6 viruses-13-02454-t006:** Distribution of high-risk HPV (HRHPV) genotypes overall and by transplantation group/immunosuppressive therapies.

	All		Transplantation Group		Immunosuppressive Therapies	
	*n*	%	*n*	%	*n*	%
Total HRHPV	29	47.5	11	50	18	46.2
Single infections	23	37.7	9	40.9	14	35.9
Multiple infections	21	34.4	8	36.4	13	33.3
HRHPV genotypes						
HPV 16	5	8.2	1	4.5	4	10.3
HPV 35	3	4.9	1	4.5	2	5.1
HPV 39	1	1.6	1	4.5	0	0
HPV 45	1	1.6	1	4.5	0	0
HPV 52	7	11.5	3	13.6	4	10.3
HPV 53	9	14.8	4	18.2	5	12.8
HPV 56	1	1.6	1	4.5	0	0
HPV 58	4	6.6	1	4.5	3	7.7
16/18	7	11.5	2	9.1	5	12.8
16/18/31/33/45/52/58	26	42.6	9	40.9	17	43.6
All women, total	61	100	22	36.1	39	63.9

**Table 7 viruses-13-02454-t007:** Distribution of high-risk HPV (HRHPV) genotypes according to the number of immunosuppressive therapies.

	All		Monotherapy		Combination Therapy	
	*n*	%	*n*	%	*n*	%
Total HRHPV	29	47.5	19	51.4	10	41.7
Single infections	23	37.7	15	40.5	8	33.3
Multiple infections	21	34.4	12	32.4	9	37.5
HRHPV genotypes						
HPV 16	5	8.2	3	8.1	2	8.3
HPV 35	3	4.9	3	8.1	0	0
HPV 39	2	3.3	1	2.7	1	4.2
HPV 45	1	1.6	1	2.7	0	0
HPV 52	9	14.8	4	10.8	5	20.8
HPV 53	9	14.8	7	18.9	2	8.3
HPV 56	1	1.6	1	2.7	0	0
HPV 58			3	8.1	2	8.3
16/18	7	11.5	4	10.8	3	12.5
16/18/31/33/45/52/58	28	45.9	17	45.9	11	45.8
All women, total	61	100	37	60.7	24	39.3

**Table 8 viruses-13-02454-t008:** Risk of HPV infection according to HIV viral load, CD4 cell count and to monotherapy/combination therapy status.

	Viral Load			CD4 Cell Count			Therapy Status		
	Negative N = 22 ^1^	Positive ^3^ N = 7 ^1^	p^2^	≤349 c/mm ^3^ N = 11 ^1^	>350 c/mm ^3^ N = 18 ^1^	*p* ^2^	Combination Therapy N = 24 ^1^	Monotherapy N = 37 ^1^	*p* ^2^
Prevalence of HPV	19 (86%)	6 (86%)	>0.9	9(82%)	16 (89%)	0.59	16 (67%)	27 (73%)	0.59
Multiple HPV infection	12 (55%)	4 (57%)	>0.9	8(73%)	8 (44%)	0.14	9 (37.5%)	12 (32.4%)	0.14
High risk HPV infection	11 (50%)	4 (57%)	>0.9	8(73%)	7 (39%)	0.08	10 (41.7%)	19 (51.4%)	0.08

^1^ n (%), ^2^
*p* = *p*-value, Fisher’s exact test, ^3^ HIV viral load quantified by Aptima HIV-1 Quant Dx test, Hologic, France. Limit of detection > 20 copies/mL (range 13.9 × 10^4^ copies/mL, mean 4.6 × 10^4^ copies/mL).

**Table 9 viruses-13-02454-t009:** Risk of HPV infection according to the type of disease (transplanted/autoimmune diseases) and to the type of immunosuppressive therapy (only patients with monotherapy have been included).

	Type of Diseases			Immunosuppressive Therapy		
	Autoimmune Diseases N = 39 ^1^	Transplanted N = 22 ^1^	*p*-Value ^2^	Glucocorticoids N = 9 ^1^	Non-Glucocorticoids N = 28 ^1^	*p*-Value ^2^
Prevalence of HPV	26 (67%)	17 (77.3%)	0.4	7 (77.8%)	20 (71.4%)	0.71
Multiple HPV infection	13 (33.3%)	8 (36.4%)	0.8	4 (44.4%)	8 (28.5%)	0.38
High risk HPV infection	18 (46.2%)	11 (50%)	0.8	4 (44.4%)	18 (64.3%)	0.29

^1^ n (%), ^2^ Pearson’s Chi-squared test. Non-glucorticoids including hydroxychloroquine, anthracyclines, dimetyl-fumarate, monoclonal antibodies, inhibitors of calcineurin, intestinal anti-inflammatory and antimetabolites.

## Data Availability

The authors confirm that the data supporting the findings of this study are available upon request to the corresponding author.

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
