# Peer review of "Prevalence and Distribution of HPV Genotypes in Immunosuppressed Patients in Lorraine Region"

_viruses, 2021, doi:10.3390/v13122454_

Round 1

Reviewer 1 Report

The aim of this work was to assess the prevalence and distribution of HPV genotypes in immunosuppressed patients, and to compare them with French cohort. Secondary objectives were to evaluate whether the risk of HPV infection was correlated with HIV viral load, CD4 cell count in HIV-infected patients and the type, number of immunosuppressive therapies, type of pathology (transplant vs autoimmune diseases) in patients undergoing long-term immunosuppressive therapy.

My comments are following:

Major points:

Line 124-140, 150 – Table 1, 2 and 3

The Authors should revise these tables. Some information may be removed as it is not further analyzed.

Line 178-186

Table 5 shows the distribution of HPV genotypes.

Whether the data relate to a single HPV infection, or whether there were co-infections with several HPV types, which seems more likely.This should be stated in the article.

 Line 188-201  Factors associated with HPV infection - it would be better to present this data in a table. It would be easier for the reader.Whether HIV RNA determinations were performed if so, what method was used.

It should then be written whether the tests were qualitative (positive, negative) or quantitative - the level of viral load.

It would be very interesting:  the relationship between type of immunosuppresive disease and prevalence of HPH genotype and between number of immunosuppresive therapies and prevalence of HPV genotype.

Author Response

I am sorry not to modify the latest version but I could not open it correctly. 

Thank you for your review.

Margot BOUDES

Reviewer 2 Report

The paper  represents a retrospective study of a stratified patient population to assess the incidence and genotyping of vaginal HPV infection.
The descriptive study is well structured and the methodological biases are correctly described (type of tests used, characteristics of the definition of immunosuppression. 

Author Response

I am sorry not to modify the latest version of the article but I could not open it correctly.

Thank you for your review.

Margot BOUDES

Round 2

Reviewer 1 Report

The Authors correct the paper in accordance with all comments. May be accepted for publications. Good luck!